

# Distribution of deep-water scleractinian and stylasterid corals across abiotic environmental gradients on three seamounts in the Anegada Passage

Steven R. Auscavitch[1], Jay J. Lunden[1], Alexandria Barkman[1], Andrea M. Quattrini[2], Amanda W.J. Demopoulos[3] and Erik E. Cordes[1]

[1] Department of Biology, Temple University, Philadelphia, PA, USA
[2] National Museum of Natural History, Smithsonian Institution, Washington, DC, USA
[3] Wetland and Aquatric Research Center, US Geological Survey, Gainesville, FL, USA

Corresponding author
Steven R. Auscavitch,
steven.auscavitch@temple.edu

## ABSTRACT

In the Caribbean Basin the distribution and diversity patterns of deep-sea scleractinian corals and stylasterid hydrocorals are poorly known compared to their shallow-water relatives. In this study, we examined species distribution and community assembly patterns of scleractinian and stylasterid corals on three high-profile seamounts within the Anegada Passage, a deep-water throughway linking the Caribbean Sea and western North Atlantic. Using remotely operated vehicle surveys conducted on the E/V *Nautilus* by the ROV *Hercules* in 2014, we characterized coral assemblages and seawater environmental variables between 162 and 2,157 m on Dog Seamount, Conrad Seamount, and Noroît Seamount. In all, 13 morphospecies of scleractinian and stylasterid corals were identified from video with stylasterids being numerically more abundant than both colonial and solitary scleractinians. Cosmopolitan framework-forming species including *Madrepora oculata* and *Solenosmilia variabilis* were present but occurred in patchy distributions among the three seamounts. Framework-forming species occurred at or above the depth of the aragonite saturation horizon with stylasterid hydrocorals being the only coral taxon observed below $\Omega_{arag}$ values of 1. Coral assemblage variation was found to be strongly associated with depth and aragonite saturation state, while other environmental variables exerted less influence. This study enhances our understanding of the factors that regulate scleractinian and stylasterid coral distribution in an underreported marginal sea and establishes a baseline for monitoring future environmental changes due to ocean acidification and deoxygenation in the tropical western Atlantic.

## INTRODUCTION

Global and regional modelling efforts in parallel with observational studies have contributed to our understanding of the distribution of framework-forming

azooxanthellate corals in the deep sea (>200 m depth). However, a significant number of data deficient localities persist, most often hindered by a lack of records from direct seafloor observation and in situ environmental data. A number of environmental variables have been observed to control the distribution of deep-water azooxanthellate scleractinian corals and stylasterid hydrocorals, including parameters linked to depth, terrain, hydrography, and seawater chemistry (*Guinotte et al., 2006*; *Cairns, 2007*; *Davies & Guinotte, 2011*). In the tropical western Atlantic, a center for deep-water scleractinian diversity (*Cairns, 2007*), additional direct observational data are needed to validate modelling efforts.

The growth of a scleractinian coral colony and formation of the aragonitic calcium carbonate skeleton are generally dependent upon the ambient seawater aragonite saturation state ($\Omega_{arag}$) being greater than 1, or supersaturated. However, some deep-water scleractinian corals, including the cosmopolitan species *Solenosmilia variabilis* and *Enallopsammia rostrata*, have been reported well below the saturation horizon (the depth at which $\Omega_{arag} = 1$) on the Tasmanian Seamounts (*Thresher et al., 2011*). A more recent survey of seamounts in the Northwestern Hawaiian Islands and Emperor Seamounts documented living colonial scleractinian reefs at $\Omega_{arag}$ as low as 0.71 (*Baco et al., 2017*). Previous studies lend support for targeted observational surveys in underexplored regions in order to delineate species distribution patterns with respect to aragonite saturation and other environmental variables. These efforts are particularly salient in light of potential and ongoing anthropogenic impacts to deep-sea coral ecosystems (*Ragnarsson et al., 2017*), including global climate change and ocean acidification (*Gleckler et al., 2016*; *Pérez et al., 2018*), deep-water drilling and resource extraction (*Cordes et al., 2016*), bottom-contact fishing (*Watling & Norse, 1998*), and deep-sea crust mining (*Miller et al., 2018*).

The Greater-Lesser Antilles transition zone is one of the most sparsely surveyed yet biogeographically important deep-water entries from the western Atlantic Ocean into the Caribbean Basin. Topographic features, such as seamounts and other submarine banks associated with deformation along tectonic plate boundaries, are important contributors to the environmental complexity of the deep-sea benthos. In the Anegada Passage, Dog Seamount, Conrad Seamount, Noroît Seamount, and Barracuda Bank are some of the most prominent of the large (>2 km in height above the surrounding seafloor) seamounts and banks present, ranging from abyssal to mesophotic depths. Typical environmental characteristics of seamounts, including substrate heterogeneity, enhanced productivity and carbon flux, and variable currents (*Rogers, 2018*), create the potential for deep-water biodiversity hotspots in the region and elevated species abundance relative to the adjacent continental margins and slopes. Previous studies indicate that at depths >200 m, azooxanthellate corals generally prefer hard substrata that exhibit topographic complexity (*Roberts et al., 2009*; *Georgian, Shedd & Cordes, 2014*). As such, seamounts are one example of topographically complex deep-water features that provide ideal hard substrate for corals (*Rogers, 1994*). Yet very few seamount benthic communities in the Caribbean region have been characterized with respect to these important abiotic gradients.

Throughout the tropical western Atlantic, scleractinian species diversity and distributions are relatively well-known from shallow waters but the extent of their

distribution below 150 m is poorly understood. For deep-water azooxanthellate corals, 81 species have been reported at depths greater than 50 m for the Greater and Lesser Antilles (*Cairns, 2007*). As with scleractinians, stylasterid hydrocorals are reportedly diverse from the Lesser Antilles (*Cairns, 1986*), can occur in high densities in the tropical western Atlantic (*Messing, Neumann & Lang, 1990*), and are composed of rigid, primarily aragonitic coralla (*Cairns & Macintyre, 1992*). Additionally, three-dimensional structures produced by deep-water stylasterids provide habitat for fishes (*Love, Lenarz & Snook, 2010*) and invertebrate fauna (*Braga-Henriques et al., 2011*; *Reed et al., 2013*). Efforts have also been made to understand the biogeographic context of Caribbean ecoregions using deep-water corals in order to support conservation strategies (*Cairns & Chapman, 2001*; *Miloslavich et al., 2010*; *Hernández-Ávila, 2014*). In the Caribbean Basin, patterns in deep-sea coral beta diversity (e.g., species turnover) between ecoregions have been attributed to topography and oceanography (*Hernández-Ávila, 2014*). Within the insular Caribbean, the Greater Antilles and Lesser Antilles regions are found to diverge into two ecoregions, one encompassing the Greater Antilles islands and the other the Lesser Antilles islands, at continental slope depths (200–2,000 m) (*Hernández-Ávila, 2014*). At larger biogeographic scales and evolutionary time, deep-water currents or water masses have been hypothesized as distribution pathways for constraining cosmopolitan habitat-forming corals like *Lophelia pertusa* in the western Atlantic (*Arantes et al., 2009*; *Henry, 2011*). While widespread seamount surveys from the Caribbean Basin remain rare, the effect of environmental variables like temperature, dissolved oxygen, and water mass has been demonstrated for demersal fishes on seamounts in the Anegada Passage (*Quattrini et al., 2017*).

The present study focuses on describing patterns in the distribution of deep-water scleractinians and stylasterid corals on three prominent seamounts in the northeastern Caribbean Sea. In addition to the primary abiotic oceanographic variables including temperature, salinity, and dissolved oxygen, we explore the distribution of aragonitic corals in the region with respect to the aragonite saturation state in this region of the western Atlantic Ocean. As a suspected driver of change in deep-water coral community structure, we also examine the relationship between water mass structure and community similarity in the bathyal zone, hypothesizing that different water masses would have distinct coral assemblages. Finally, we aim to identify abiotic oceanographic variables that most strongly influence the presence of hard corals and their contribution to changes in community similarity for this locality.

## MATERIALS AND METHODS

### ROV surveys

This study examined sites within the Anegada Passage in the northeastern corner of the Caribbean Sea (Fig. 1). Three seamounts of varying summit depths were surveyed in September 2014 using the ROV *Hercules* during E/V *Nautilus* cruise NA052 (Table 1). Three dives were conducted on Dog Seamount (276–1,035 m depth), two were conducted on Conrad Seamount (162–1,314 m depth), and two were conducted on Noroît Seamount

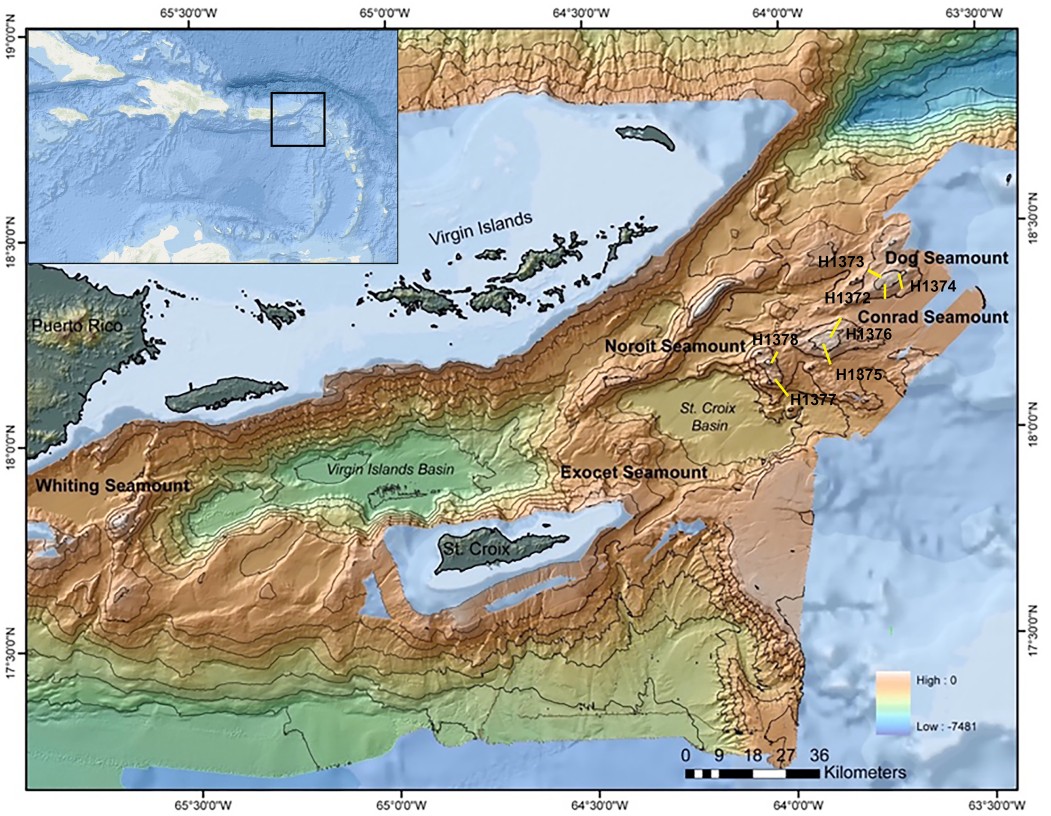

**Figure 1 Multibeam bathymetric map of the Anegada Passage seamounts.** The locations of Dog, Conrad and Noroît Seamount are indicated in bold. Dive locations are overlaid in yellow and labeled by dive number. The bathymetry color ramp in the lower right indicates depth values measured in meters. Transect segment lengths are not drawn to scale.

(949–2,206 m depth). Efforts were made to cover similar depth ranges on each feature for direct comparison as permitted by the bathymetry.

The ROV was deployed to a maximum target depth on each seamount and generally moved to shallower depths up slope. The ROV continuously traversed the seafloor as near to the bottom as practical at a slow, steady speed (~0.1–0.2 knots, ~0.1 m/s); however, transects were occasionally interrupted by stopping the ROV for sampling and detailed camera zooms. The ROV was equipped with a high-definition camera and paired scaling lasers (10 cm apart). During the dives, the forward-facing cameras were set on wide-angle view, but frequent snap-zooms (up to 20 s) were conducted to aid in species identification. The ROV was also equipped with a Seabird FastCAT 49 conductivity-temperature-depth (CTD) logger and an Aanderaa oxygen optode to measure dissolved oxygen (DO). The ROV was tracked on the seafloor using an ultra-short baseline (USBL) tracking system as well as a Doppler Velocity Log (DVLNAV).

### Seawater collection and carbonate chemistry analysis

Seawater samples ($n = 34$) were collected both in the water column and at the benthos using Niskin bottles mounted to the ROV *Hercules* (Table S1). On bottom, seawater
**Table 1 ROV dive transect details for 2014 surveys of the Anegada Passage seamounts.** Start and end coordinates for each transect, time spent in visual contact with the seafloor, and number of water samples collected at depth over the dive interval are indicated.

| Dive | Seamount | Start coordinates | End coordinates | Depth range (m) | Number of Niskin bottle samples | Total bottom time (hh:mm) |
|---|---|---|---|---|---|---|
| H1372 | Dog | 18°18.7385 N | 18°19.6177 N | 717–1,035 | 6 | 07:56 |
| | | 63°46.0896 W | 63°46.1645 W | | | |
| H1373 | Dog | 18°22.6480 N | 18°21.9711 N | 503–784 | 6 | 11:09 |
| | | 63°46.1953 W | 63°45.9306 W | | | |
| H1374 | Dog | 18°20.4464 N | 18°21.9311 N | 276–601 | 5 | 09:58 |
| | | 63°43.6091 W | 63°43.5024 W | | | |
| H1375 | Conrad | 18°10.5420 N | 18°13.0029 N | 162–876 | 5 | 31:40 |
| | | 63°53.1020 W | 63°53.3459 W | | | |
| H1376 | Conrad | 18°16.4904 N | 18°14.2223 N | 344–1,267 | 6 | 22:38 |
| | | 63°52.3364 W | 63°52.9767 W | | | |
| H1377 | Noroît | 18°05.4657 N | 18° 07.0695 N | 881–2,157 | 6 | 17:10 |
| | | 64°00.1741 W | 64° 01.1760 W | | | |
| H1378 | Noroît | 18°09.6969 N | 18° 09.4122 N | 949–1,035 | 2 | 06:01 |
| | | 64°01.3606 W | 64° 01.8888 W | | | |

samples were collected within 1–2 m of the benthos and usually co-occurred with the observation of scleractinian colonies on the seafloor. For all seawater samples, co-located physical data (pressure, temperature, salinity and oxygen) were obtained using vehicle mounted conductivity-temperature-depth (SBE 49Plus; Paroscientific Digiquartz, Redmond, WA, USA) and oxygen (Aanderaa Optode 3830) sensors, respectively. Upon recovery of the ROV, samples were immediately transferred to 500 mL high-density polyethylene (HDPE) containers according to Best Practices for Ocean $CO_2$ measurements (*Dickson, Sabine & Christian, 2007*). While HDPE containers are suitable for long-term storage of seawater for total alkalinity analyses (*Huang, Wang & Cai, 2012*), they are not suitable for long-term storage for pH or dissolved inorganic carbon analyses as they are permeable to $CO_2$. Accordingly, pH was measured within 1 h of collection onboard the vessel. Immediately following pH measurement, samples were poisoned with 100 µL saturated mercuric chloride solution and stored in a cool, dark location. Total alkalinity was measured in the laboratory in triplicate according to methods previously described (*Lunden, Georgian & Cordes, 2013*; *Georgian et al., 2016a*), and final $\Omega_{arag}$ values were computed using CO2calc (*Robbins et al., 2010*). Due to the logistical challenge of pairing a discrete water sample with each individual scleractinian coral observation, we generated a predictive model to interpolate $\Omega_{arag}$ at the depth of each coral observation based on temperature, dissolved oxygen, and salinity (for methodology see *Georgian et al., 2016a*).

## ROV video analysis

Colonial and solitary coral occurrences were documented using high-definition video transects during each of the ROV dives. Video segments where collections occurred, where
the vehicle was too high off the bottom, moving backwards, or of generally poor quality, were removed from analysis. For the purpose of this analysis, we defined structure-forming corals as colonial members of the Scleractinia and hydrozoan family Stylasteridae. Consistent morphospecies identifications were used throughout the analyses. Voucher specimens for morphological identification were obtained using the ROV platform. If necessary, further identifications of coral species were made using taxonomic keys and assistance from taxonomic specialists. Colony height was measured, where possible, by referencing scaling lasers. Individuals or colonies above a 5-cm height threshold were typically identifiable based on laser scalers. If occurrences could not be readily identified below the 5-cm threshold, they were omitted from analysis. Each observation was paired with an associated time-stamp for reference to in situ environmental data (temperature, depth, dissolved oxygen concentration, and salinity) collected by ROV CTD and oxygen optode sensors.

## Community analyses

Sampling effort was evaluated for seamounts individually and for all seamounts combined by sample-based species accumulation curves. In order to assess community relationships among hard-coral assemblages, species abundance values (number of colonies or individuals for each species) were binned into 100 m depth segments for each seamount transect resulting in 28 total samples. Community-level analyses were conducted using standardized and square root transformed species abundance data in PRIMER v7 with PERMANOVA add-on (*Clarke & Gorley, 2006*; *Anderson, Gorley & Clarke, 2008*). Transformed and standardized species abundances were compiled into a Bray–Curtis resemblance matrix for further analyses. In order to test for significant differences among hard coral assemblages on different seamounts, a one-way analysis of similarity (ANOSIM) was conducted between features as well as between local water masses. Non-metric multidimensional scaling ordinations (nMDS) were conducted across all 28 samples. nMDS plots were overlaid with similarity profile (SIMPROF) analysis to show significant groupings of samples at the 95% level or greater (*Clarke, Somerfield & Gorley, 2008*). Similarity percentage (SIMPER) tests were used to identify taxa that contributed disproportionately to assemblage similarities within and between seamounts and water masses.

Multivariate analyses were also used to explore the abiotic variables relevant to the variation observed in the coral assemblages. Environmental data for temperature, salinity, dissolved oxygen, and $\Omega_{arag}$ were obtained from the calculated mean in each 100 m depth segment within every transect. The mean was then 4th-root transformed and normalized within each variable. The BEST (BIO-ENV) routine (*Clarke, Somerfield & Gorley, 2008*) was applied to the dataset to determine which environmental variables would be useful predictor variables. A distance-based linear model (DistLM) with Akaike information criterion (AIC) was applied using the PERMANOVA add-on in PRIMER v7 (*Anderson, Gorley & Clarke, 2008*). Visualizations of the resemblance matrix with predictor variables were observed using a dbRDA (distance-based redundancy analysis) ordination with DistLM overlay.

 

**Table 2 Summary of records and environmental variables for each taxon.** Ranges of values for number of observations, depth, temperature, salinity, dissolved oxygen, model predicted $\Omega_{Arag}$, and colony height are reported for each species by taxonomic grouping.

| Class | Family | Species | Number of observations | Depth (m) | Temperature (°C) | Salinity (psu) | Dissolved oxygen (mg/L) | Model predicted $\Omega_{arag}$ | Height (cm) |
|---|---|---|---|---|---|---|---|---|---|
| Hydrozoa | Stylasteridae | *Stylaster* sp. 1 | 65 | 166–174 | 22.1–22.5 | 37.00–37.05 | 7–7.12 | 3.38–3.45 | – |
| | | *Crypthelia* sp. 1 | 49 | 304–1,096 | 4.9–18.0 | 34.53–36.55 | 2.41–7.22 | 0.99–2.83 | 1–12 |
| | | *Stylaster* cf. *duchassaingi* | 30 | 181–408 | 13.8–22.1 | 35.82–37.01 | 5.39–6.99 | 2.07–3.39 | 12–37 |
| | | Stylasteridae spp. | 2 | 525–838 | 6.7–10.9 | 34.89–35.39 | 4.96–5.28 | 1.08–1.62 | 3–5 |
| Anthozoa | Caryophylliidae | *Solenosmilia variabilis* | 44 | 490–569 | 10.9–13.3 | 35.37–35.75 | 4.88–5.15 | 1.61–1.98 | 12–18 |
| | | *Caryophyllia* sp. 1 | 3 | 665–677 | 8.2–8.7 | 35.03–35.07 | 4.78–4.86 | 1.24–1.29 | – |
| | Oculinidae | *Madrepora oculata* | 22 | 784–1,540 | 4.3–6.9 | 34.53–34.97 | 5.22–8.23 | 1.00–1.10 | 10–125 |
| | Pocilloporidae | *Madracis myriaster* | 22 | 253–311 | 17.6–18.7 | 36.49–36.65 | 6.83–7.01 | 2.78–2.92 | 6–35 |
| | *Incertae familiae* | Scleractinia spp. (solitary) | 11 | 311–849 | 6.5–17.8 | 34.83–36.51 | 4.76–6.98 | 1.07–2.8 | 4–5 |
| | Dendrophylliidae | *Dendrophyllia alternata* | 5 | 490–598 | 10.7–13 | 35.33–35.71 | 4.84–5.13 | 1.57–1.95 | 16 |
| | | *Enallopsammia rostrata* | 2 | 785 | 7.0 | 34.921 | 5.2 | 1.11 | 20–75 |
| | | Dendrophylliidae spp. | 1 | 437 | 13.6 | 35.794 | 5.29 | 2.03 | 5 |
| | Flabellidae | *Javania cailleti* | 8 | 518–1,626 | 4.2–12.0 | 34.557–35.54 | 4.96–8.34 | 1.02–1.77 | 4–7 |

## RESULTS

### Survey summaries

A total of 106 h of bottom time was assessed across seven dives (Table 1). Video records were annotated between the depth range of 162–2,157 m. In all, 264 observations of solitary and framework-forming scleractinian corals and stylasterid hydrocorals were made across all three seamounts (Table 2). Three dives at Dog Seamount yielded 64 records from 284 to 849 m. From Conrad Seamount, 182 records were made from 166 to 1,230 m across two dives. The deepest two dives at Noroît Seamount had 18 records reported from 1,014 to 1,626 m. Stylasterids made up the majority (53%) of coral observations, however most were observed at depths shallower than 400 m (Table 2). The majority of scleractinians observed were colonial, framework-forming species (100 colonies observed) while only 22 observations were made of solitary coral species, as they were often on or below the threshold for identification. For this reason, abundances of solitary scleractinians represent a minimum value. No scleractinians or stylasterids above the 5 cm size threshold were observed deeper than 1,626 m on any seamount.

### Water mass structure

Downcast CTD profiles from the ROV sensors were plotted and assessed at each seamount using Ocean Data View v5 (*Schlitzer, 2019*). Profiles for each seamount were combined to

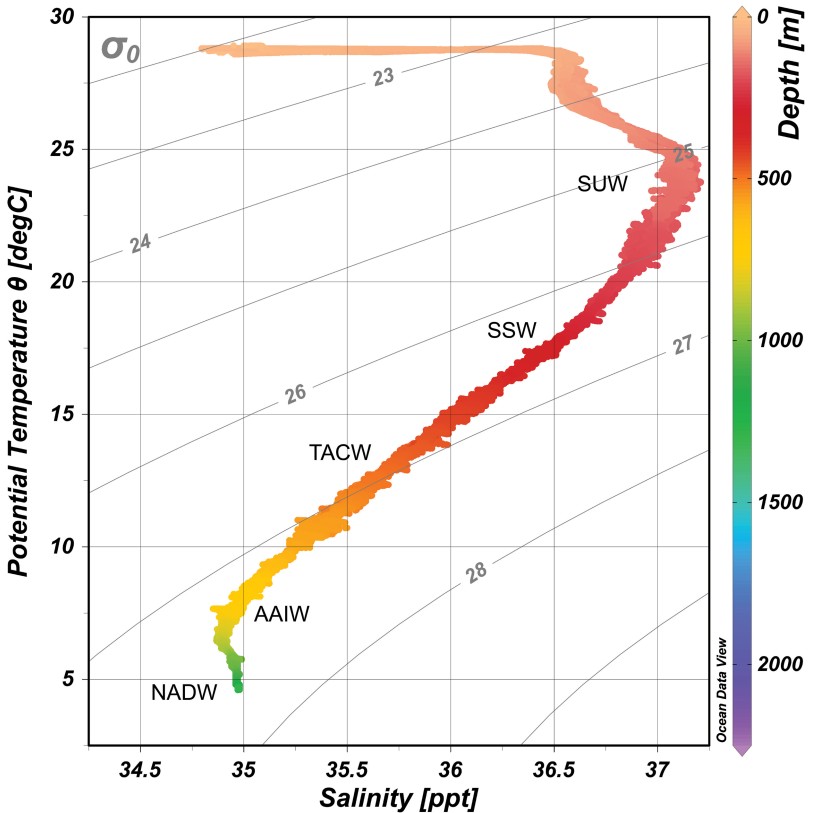

**Figure 2 Temperature-Salinity plot for the Anegada Passage based on CTD water column profiles.**
Water masses are overlaid and abbreviated by the following: SUW, Subtropical underwater; SSW, Sargasso Sea Water; TACW, Tropical Atlantic Central Water; AAIW, Antarctic Intermediate Water; NADW, North Atlantic Deep Water. Isopycnal surfaces ($\sigma_\theta$) are indicated in gray.

create one consensus profile for the Anegada Passage (Fig. 2). Water masses were identified following published records of temperature, salinity, and dissolved oxygen profiles for the northeast Caribbean and Anegada Passage; these include Subtropical Underwater (SUW) 100–200 m, Sargasso Sea Water (SSW) 200–400 m, Tropical Atlantic Central Water (TACW) 400–700 m, Antarctic Intermediate Water (AAIW) 700–1,200 m, and North Atlantic Deep Water (NADW) (>1,200 m) (*Morrison & Nowlin, 1982*).

## Carbonate chemistry analysis

Water samples for carbonate chemistry analyses were collected at depth from 50 to 2,170 m. Total alkalinity ranged from 2,291.4 to 2,405.9 µmol·kg$^{-1}$, and pH ranged from 7.83 to 8.11, with minimum pH observed at 795 m depth. Measured $\Omega_{arag}$ values from Niskin bottle collections ranged from 4.13 at 50 m depth to 0.99 at 2,170 m depth. The aragonite saturation state of the water column was modelled according to the following equation: $\Omega_{arag} = (T \times 0.11407018) + (O \times 0.00302922) + (S \times 0.18168448) - 6.5216044$ (stepwise backward regression, $R^2 = 0.9857$, $p < 0.001$) where $T$ = temperature in °C, $O$ = oxygen concentration in µmol·L$^{-1}$, and $S$ = salinity in parts per thousand (ppt).

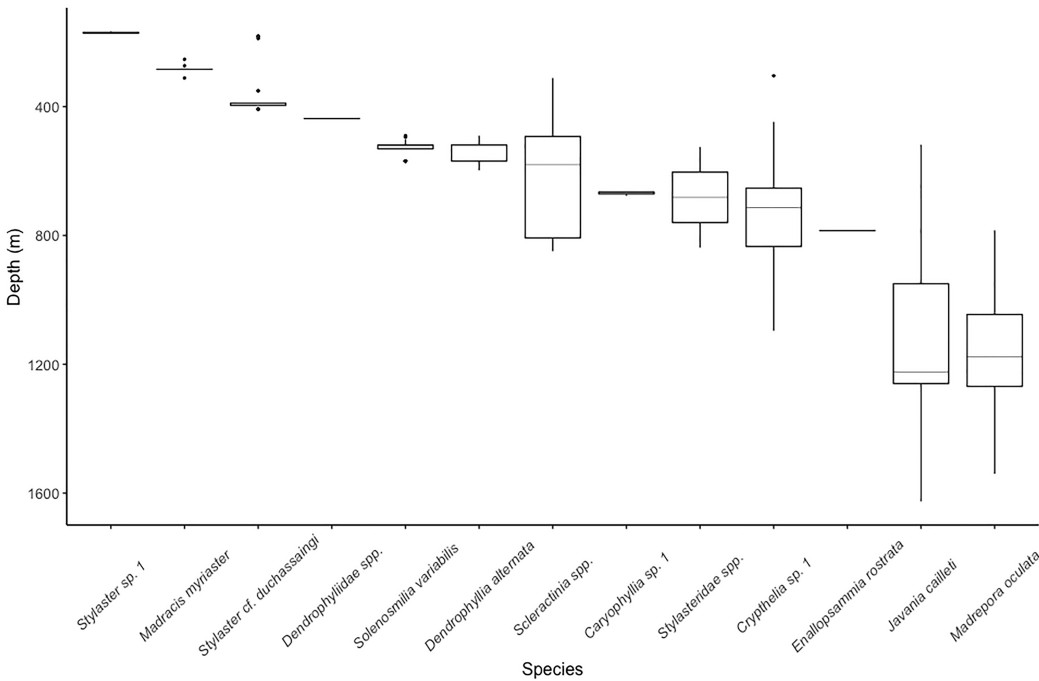

**Figure 3 Depth distribution for scleractinian and stylasterid corals occurring on Dog, Conrad, and Noroît Seamounts.** Box plots represent the interquartile range with whiskers extending from the upper and lower quartiles to the minimum and maximum values. Horizontal lines represent the median depth. Individual points represent outliers. Species are arranged on the *x*-axis from shallowest to deepest by median depth of occurrence.

From this, predicted $\Omega_{arag}$ values ranged from 4.11 to 1.05, with highest value at 51 m at Dog Seamount and lowest value at 2,195 m at Noroît Seamount.

## Coral species distribution patterns

The single most abundant scleractinian coral species was *Solenosmilia variabilis* (44 colonies), but it was only observed on Conrad Seamount in a relatively narrow depth range, 409–569 m. Many structure-forming colonies were found to be associated with boulder and low, outcropping, hard rock substrate. This was followed by the more widespread *Madrepora oculata* (22 colonies), which occurred between 784 and 1,540 m on all seamounts (Fig. 3). *Madrepora oculata*, which displayed two different growth forms from thin and sparsely branching to thick, robust colonies, possessed the widest depth distribution range for any colonial scleractinian coral. The deepest solitary scleractinian coral, *Javania cailleti*, was observed at 1,626 m; this occurrence corresponds with one of the lowest measured $\Omega_{arag}$ values of 1.01.

The upper bathyal depths, shallower than 700 m, contained species with narrower depth distributions than those at lower bathyal depths (>800 m) (Fig. 3). The sub-700 m assemblage was composed of three species of scleractinians (*E. rostrata*, *M. oculata*, and *J. cailleti*) and one stylasterid (*Crypthelia* sp. 1). The shallowest depths, usually coinciding with the seamount summit, were dominated by two species of stylasterids, *Stylaster* sp. 1 and *Stylaster* cf. *duchassaingi*, as well as one azooxanthellate scleractinian,

**Table 3 Summary of measured values of $\Omega_{\text{Arag}}$ from water samples taken directly adjacent to corals.** Sampling events are reported for each species arranged by taxonomic grouping.

| Class | Family | Species | Number of adjacent water samples | $\Omega_{\text{arag}}$ measured |
|---|---|---|---|---|
| Hydrozoa | Stylasteridae | *Stylaster* sp. 1 | 0 | – |
| | | *Crypthelia* sp. 1 | 0 | – |
| | | *Stylaster* cf. *duchassaingi* | 0 | – |
| | | Stylasteridae spp. | 0 | – |
| Anthozoa | Caryophylliidae | *Solenosmilia variabilis* | 1 | 1.63 |
| | | *Caryophyllia* sp. 1 | 1 | 1.24 |
| | Oculinidae | *Madrepora oculata* | 2 | 1.13–1.16 |
| | Pocilloporidae | *Madracis myriaster* | 1 | 2.7 |
| | *Incertae familiae* | Scleractinia spp. (solitary) | 0 | – |
| | Dendrophylliidae | *Dendrophyllia alternata* | 1 | 1.35 |
| | | *Enallopsammia rostrata* | 1 | 1.21 |
| | | Dendrophylliidae spp. | 1 | 1.82 |
| | Flabellidae | *Javania cailleti* | 1 | 1.16 |

*Madracis myriaster*. On Conrad Seamount, pink and purple coralline algal crusts were observed as deep as 256 m depth and continued to be observed through shallower depths dominated by sponges, stylasterids, and black corals.

The majority of stylasterid species were restricted to depths less than 400 m with only *Crypthelia* exhibiting a wider depth distribution (Fig. 3). Only three morphospecies of stylasterids were large enough to be consistently identified during video annotation. Other unidentifiable stylasterids were combined into a fourth group, Stylasteridae spp. and shown for reference. The most abundant species, *Stylaster* sp. 1, was found in a narrow depth range between 166 and 174 m on the summit of Conrad Seamount.

## Patterns of coral occurrences with aragonite saturation state

Measured $\Omega_{\text{arag}}$ values from water collected adjacent to corals were not significantly different (*t*-test, $N = 9$, $t = -0.23341$, $p = 0.818$) compared to what was calculated using the modelled aragonite saturation state data (Fig. S1). All paired water samplings adjacent to coral observations occurred at or above the aragonite saturation horizon ($\Omega_{\text{arag}} > 1$) (Table 3). Multiple water samples for a single species were collected for only *M. oculata*. A full table of all 34 Niskin water bottle measurements and matched environmental variables is provided (Table S1).

Using predicted values derived from CTD and sensor data on temperature, oxygen concentration, and salinity, all scleractinian and stylasterid corals were observed at $\Omega_{\text{arag}}$ values of 0.99–3.45 (Fig. 4). For scleractinians, *Javania cailleti* occurred across the largest aragonite saturation state range, between 1.01 and 1.77. Among the Stylasteridae, *Crypthelia* sp. 1 occurred across the greatest range of saturation states from as low as

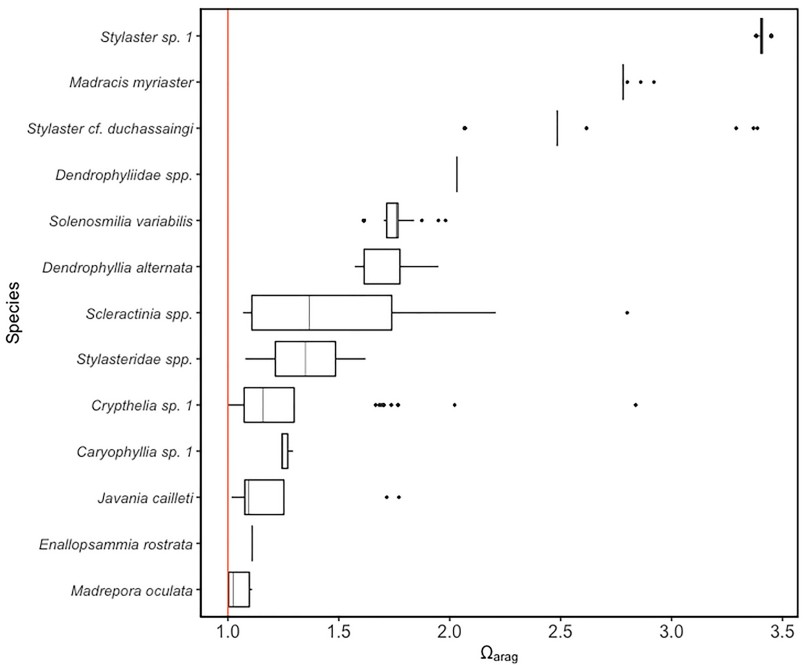

**Figure 4 Distribution of scleractinian and stylasterid species against model predicted $\Omega_{Arag}$ values on the Anegada Passage seamounts.** Box plots represent the interquartile range with whiskers extending from the upper and lower quartiles to the minimum and maximum values. Vertical lines represent the median aragonite saturation state. Individual points represent outliers. Species are arranged on the $y$-axis from highest to lowest median aragonite saturation occurrence. A solid red vertical line indicates the aragonite saturation horizon, where $\Omega_{Arag} = 1$.

0.99 up to 2.83. Only three species were observed at or around the aragonite saturation horizon based on predicted saturation state values, *Crypthelia* sp. 1 ($\Omega_{arag} = 0.99$), *M. oculata* ($\Omega_{arag} = 1.00$) and *J. cailleti* ($\Omega_{arag} = 1.01$).

## Community structure patterns

Species-accumulation curves were evaluated for seamounts individually and in aggregate for all seamounts. Combined, all seamounts (100–1,700 m) revealed a more complete sampling effort than among single seamount coral assemblages (Fig. 5). Individually, species accumulation curves for Dog, Conrad, and Noroît seamounts were not asymptotic, and therefore are likely to accumulate additional coral species with increased sampling effort.

Analysis of similarity (two-way nested, depth within seamount, Global $R = 0.122$, $p = 0.05$) conducted among seamounts revealed significant differences in coral assemblages between Dog and Noroît Seamounts ($R = 0.544$, $p = 0.006$), but not Conrad and Dog seamounts ($R = -0.001$, $p = 0.379$) or Conrad and Noroît seamounts ($R = 0.029$, $p = 0.292$) (Table S2). A one-way ANOSIM (Global $R = 0.464$, $p = 0.001$) between water masses revealed significant differences in coral assemblages between SSW and TACW ($R = 0.46$, $p = 0.003$), AAIW ($R = 0.53$, $p = 0.002$), and NADW ($R = 0.60$, $p = 0.008$) as well as between TACW and NADW ($R = 0.93$, $p = 0.001$) and AAIW and NADW ($R = 0.361$, $p = 0.01$) (Table S3).

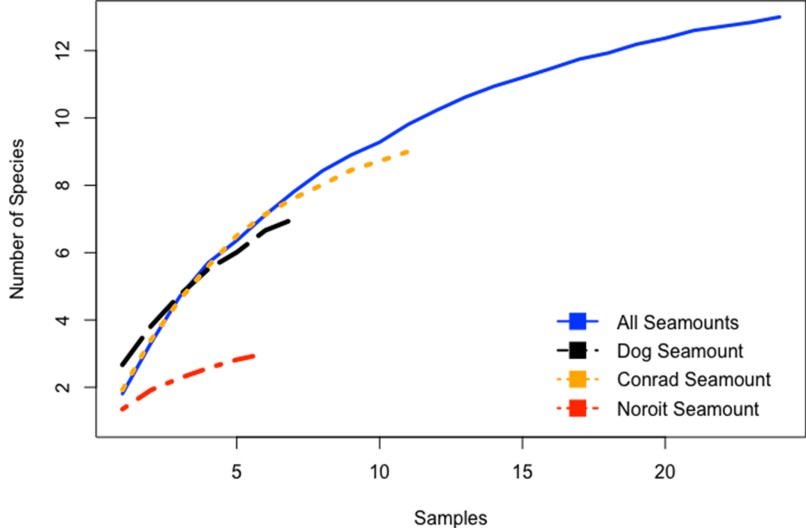

**Figure 5 Species accumulation curves for the Anegada Passage seamounts.** Curves are plotted for all transects on Dog Seamount (276–1,035 m), Conrad Seamount (162–1,267 m), Noroit Seamount (881–2,157 m), as well as for all three seamounts combined (162–2,157 m).

Non-metric multi-dimensional ordination with SIMPROF groupings revealed five statistically significant groupings; two shallow assemblages composed of 100 m depth-binned samples from between 100–600 m and 200–400 m, two mid-depth assemblages (500–700 m and 700–1,100 m), and one deep assemblage (1,100–1,600 m) (Fig. 6). Outliers from these groupings occurred in the 600–700 m and 1,100–1,200 m depth bins on Conrad Seamount and between 1,300 and 1,400 m on Noroît Seamount. Within SIMPROF groupings, the shallowest depth grouping showed the greatest amount of dissimilarity among samples (a metric of beta diversity) among the three seamounts while the lowest occurred in the deepest group.

Dog Seamount exhibited the lowest beta diversity of coral assemblages among all three seamounts and the highest average similarity at 61% (one-way SIMPER analysis), with the average similarity between 100 m depth bins being most strongly influenced by *Crypthelia* sp. 1, which contributed to 96.6% of the relative abundance. Noroît Seamount had the second highest similarity (38.2%), with *M. oculata* accounting for 84.7% of the similarity. Conrad Seamount had the lowest average similarity (15%), with stylasterids *Crypthelia* sp. 1 and *Stylaster* cf. *duchassaingi* being the greatest contributors to average similarity with 48.4% and 21.5%, respectively. Between seamounts, Noroît differed from Dog and Conrad, primarily due to the higher abundance of *M. oculata* at Noroît Seamount. Noroît Seamount had nearly triple the average abundance of *M. oculata* compared to Conrad Seamount. Conrad and Dog Seamounts differed primarily due to the contribution of *Crypthelia* sp. 1 and *M. myriaster*, which were present on Conrad at 2–3 times higher average abundance than at Dog Seamount.
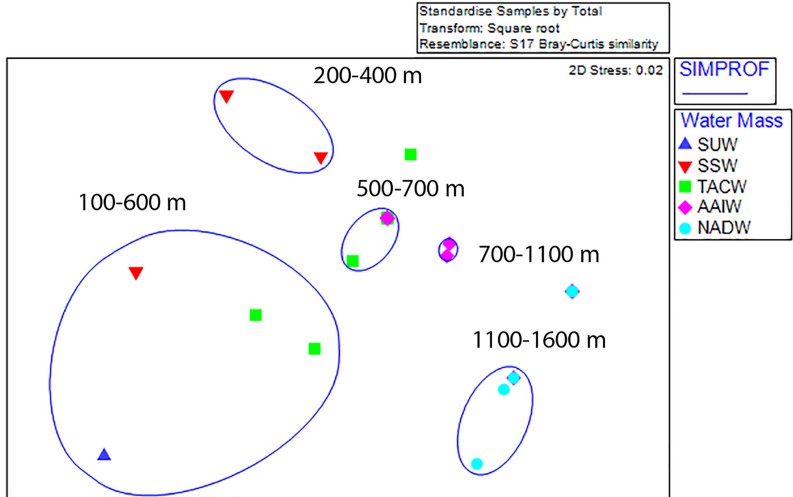

**Figure 6 Non-metric multidimensional scaling analysis of standardized, square-root transformed coral assemblages on Dog, Conrad, and Noroît Seamounts.** SIMPROF groups are overlaid around significant groupings at the 95% level or above. Depth ranges displayed within each statistical grouping are indicated in black text. Water masses in the legend are abbreviated by the following: Subtropical underwater (SUW), Sargasso Sea Water (SSW), Tropical Atlantic Central Water (TACW), Antarctic Intermediate Water (AAIW), and North Atlantic Deep Water (NADW).

Within water masses, the greatest average similarity of coral assemblages occurred within NADW (54%) followed by TACW (51%) and finally SSW at (32%). The abundance of scleractinians was responsible for greater similarities within deep water masses (*M. oculata* and *Javania cailleti* in NADW) while stylasterids were more commonly associated with driving patterns of similarity within intermediate and shallower water masses (*Crypthelia* sp. 1 in TACW, AAIW and *Stylaster* cf. *duchassaingi* in SSW). Between immediately adjacent water masses, the greatest average dissimilarity was observed between SUW and SSW (92.3%) and the lowest between TACW and AAIW (56.7%), indicating higher rates of turnover (beta diversity) for shallower water masses than deeper ones.

Results from the BEST analysis indicated that the largest percentage of biological variation was correlated with depth ($r = 0.536$). The BEST routine also indicated that the greatest correlation occurred with the four combined factors of depth, $\Omega_{arag}$, temperature, and dissolved oxygen ($r = 0.614$). Sequential tests in the DistLM analysis indicated that depth (AIC = 223.88, $p = 0.001$) and $\Omega_{arag}$ (AIC = 215.87, $p = 0.001$) were the greatest explanatory variables influencing coral assemblages on seamounts in the Anegada Passage. Temperature, salinity, and oxygen did not explain a significant portion of the biological variation in this test. Redundancy analyses resulted in 59.4% of the DistLM model variation explained by the primary axis (dbRDA1) and 32.6% explained by the second (dbRDA2) (Fig. 7). Likewise, the first axis explained 32.8% of the total biological variation observed and the second axis explained 18%. The primary axis was most closely correlated with oxygen ($r = -0.82$), while the second was most closely related to temperature ($r = -0.61$).

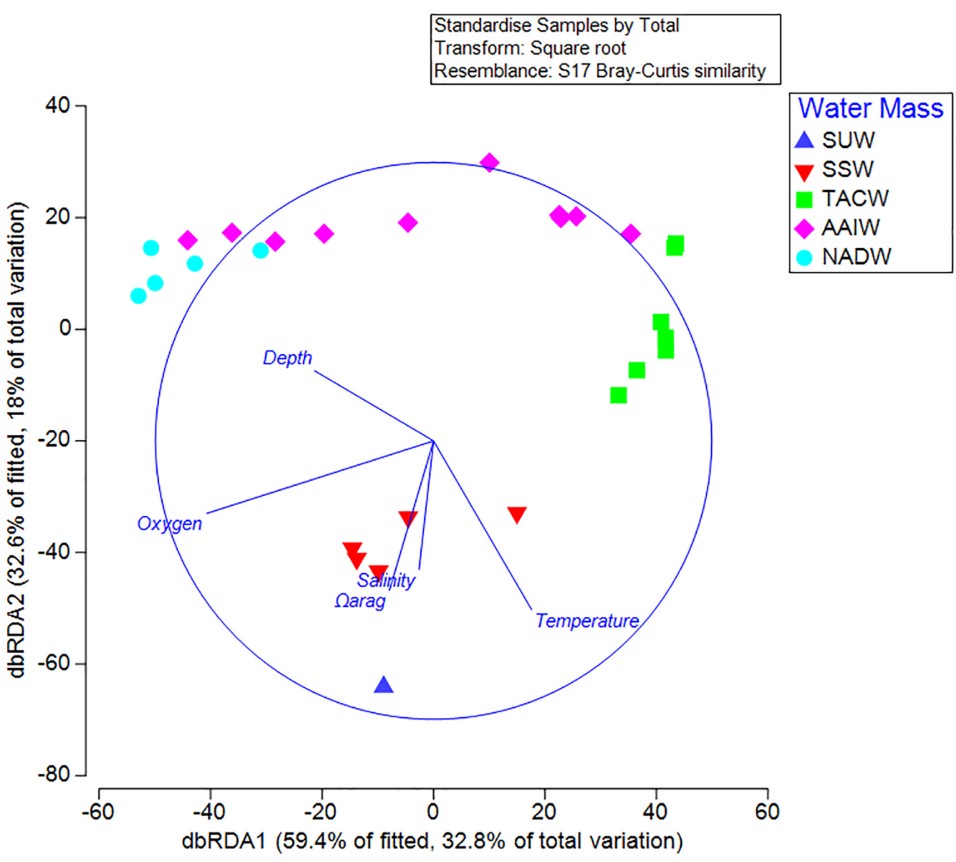

**Figure 7 Distance-based linear model and redundancy analysis of coral assemblages and oceanographic variables.** Sample points represent one 100 m depth assemblage. Axes shown are the results of a distance-based redundancy analysis with percent variation explained by the first and second axes. Water masses in the legend are abbreviated by the following: Subtropical underwater (SUW), Sargasso Sea Water (SSW), Tropical Atlantic Central Water (TACW), Antarctic Intermediate Water (AAIW), and North Atlantic Deep Water (NADW).

## DISCUSSION

Seamounts in the Anegada Passage were found to harbor communities of stony and lace corals throughout the bathyal depth range from summit depths as shallow as 166 m to a maximum of 1,626 m. Both stylasterid and scleractinian coral species with aragonitic skeletons were largely present above the aragonite saturation horizon in this region of the western Atlantic Ocean, with the understanding that sampling effort below the ASH was limited to one transect at Noroît Seamount (Table 1). Nevertheless, we were able to identify the depth of the ASH as occurring between 2,000 and 2,200 m in the Anegada Passage (Fig. S1), which is consistent with previous reports from the North Atlantic Ocean (*Jiang et al., 2015*). Only stylasterids in the genus *Crypthelia* were observed to occur below the aragonite saturation horizon ($\Omega_{arag} = 0.99$), but others, including the framework-forming species *Madrepora oculata*, occurred at saturation states as low as $\Omega_{arag} = 1.0$. Given the present findings and that aragonitic coral species have been observed at depths below that of the ASH (>2,000 m) in other parts of the Caribbean Basin (*Cairns, 1979*), it is still reasonable to expect that aragonitic corals can persist below the

aragonite saturation horizon within the Anegada Passage. Also noteworthy was the observation of live crustose coralline algae at a depth of 256 m, which is comparable to the deepest known record from San Salvador Island (268 m) in The Bahamas (*Littler et al., 1986*).

Overlying water masses were a significant indicator of community assembly differences between SSW (Sargasso Sea Water) and deeper water masses, but not the shallowest water mass, Subtropical Underwater (SUW) and deeper strata of the water column. Oceanographic variables that most strongly influenced the presence of aragonitic corals and their contribution to changes in community similarity were depth and aragonite saturation state, with temperature, salinity, and dissolved oxygen making less significant contributions.

The tropical western Atlantic is a known diversity center for azooxanthellate scleractinian corals and stylasterid hydrocorals (*Cairns, 2007*, *2011*). This study provides new distribution records for deep-water coral species in the northeastern Caribbean and builds on existing efforts to characterize the azooxanthellate coral fauna of the tropical western Atlantic (*Cairns, 1979*; *Cairns & Chapman, 2001*; *Lutz & Ginsburg, 2007*). Specifically, seven species or morphospecies of stony corals within the genera *Javania*, *Madrepora*, *Madracis*, *Enallopsammia*, *Dendrophyllia*, and *Caryophyllia* have been added to local species inventories in the Anegada Passage (Fig. 8). New records from photographic and physical specimens aid in resolving the complex biogeography of the Greater-Lesser Antilles Transition zone seamounts with respect to the western North Atlantic (Table S4). The paucity of records paired with environmental parameters from global biogeographic databases makes these observations critical to understanding the species distribution dynamics of marginal Atlantic seas and how coral distribution may be affected by future ocean climatic changes.

Stony coral assemblages in the Anegada Passage were similar to those observed in other parts of the western Atlantic Ocean with a few noteworthy absences. Absent from our records from the Anegada Passage seamounts include reef-forming species like *L. pertusa* and *E. profunda*, as well as cosmopolitan cup coral species like *Desmophyllum dianthus*, which are reportedly more common throughout the U.S. southeast continental shelf and Gulf of Mexico (*Schroeder et al., 2005*; *Reed, Weaver & Pomponi, 2006*; *Brooke & Schroeder, 2007*; *Lunden, Georgian & Cordes, 2013*; *Georgian et al., 2016a*). While fossilized *Lophelia* rubble have been reported in the Colombian Caribbean (*Santodomingo et al., 2007*) and live *Lophelia* has been observed on the Brazilian continental margin (*Arantes et al., 2009*), the southern Caribbean (*Hernández-Ávila, 2014*), and off Roatan, Honduras (*Henry, 2011*), members of this genus have not been extensively reported in the insular Caribbean (*Ocean Biogeographic Information System (OBIS), 2019*).

We were also able to identify some variability in local distribution patterns among seamounts for three colonial scleractinian species. *Enallopsammia rostrata*, *Dendrophyllia alternata*, and *Solenosmilia variabilis* appear to have patchy distributions on the Anegada Passage seamounts. For example, only two colonies of *E. rostrata* were observed, both on Dog Seamount at 785 m, and only five colonies of *D. alternata* were observed on Conrad Seamount between 490 and 598 m (Fig. 3). Similarly, *S. variabilis* was only

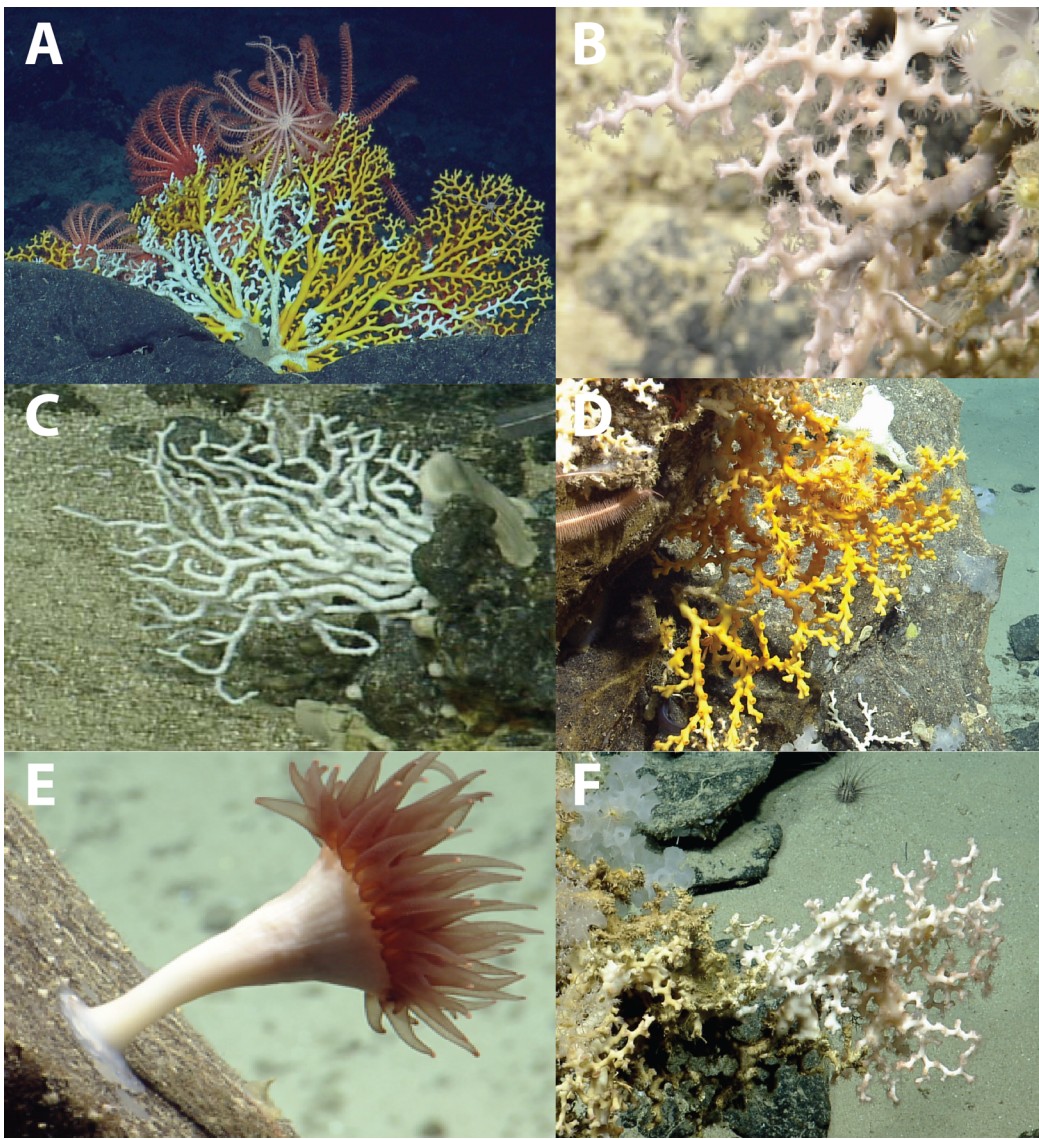

**Figure 8** **Deep-water scleractinians from the Anegada Passage seamounts.** (A) *Enallopsammia rostrata*. (B) *Madrepora oculata*. (C) *Madracis myriaster*. (D) *Dendrophyllia alternata*. (E) *Javania cailleti*. (F) *Solenosmilia variabilis*.

observed in patches on Conrad Seamount from 490 to 569 m (Fig. 3). *Solenosmilia* has not been widely reported from the greater Caribbean basin based on records from the Ocean Biogeographic Information System (*Ocean Biogeographic Information System (OBIS), 2019*), but is known to be a contributor to coral mound formation off Brazil (*Raddatz et al., 2020*). *Solenosmilia* has been more commonly reported to asexually reproduce, and thus has a relatively short dispersal ability (*Miller & Gunasekera, 2017*), potentially explaining its patchy distribution.

Several coral taxa encountered in the Anegada Passage are important species for understanding biogeographic patterns of the Caribbean bathyal zone. *Madracis myriaster* was observed both on Dog and Conrad Seamounts, but only between 253 and 311 m.

This depth range is also similar to that observed in the western North Atlantic off Bermuda (90–300 m) (*Stefanoudis et al., 2019*). In the Caribbean, *M. myriaster* has been observed at similar depths and in high densities during surveys by the ROV *Deep Discoverer* on the NOAA Ship *Okeanos Explorer* east of Vieques Island and in the Mona Passage (*Wagner et al., 2019*). Northern and eastern Caribbean coral communities at continental shelf depths have been found to be dissimilar from the southern Caribbean, such that they may constitute distinct ecoregions (*Hernández-Ávila, 2014*). These differences were found to be driven by differences in the abundance of *M. myriaster* (Fig. 8C), which is more common at mesophotic and upper bathyal depths (*Reyes et al., 2005*; *Santodomingo et al., 2007*; *Hernández-Ávila, 2014*; *Stefanoudis et al., 2019*). At lower bathyal depths, regional biogeographic differences have also been attributed to variation in the frequency of solitary scleractinian species (e.g., *Stephanocyathus* spp., *Fungiacyathus* sp.) that may be difficult to detect during video transects (*Hernández-Ávila, 2014*). A greater diversity of solitary scleractinian corals and smaller stylasterids may have been present in the video but could not be documented due to limitations of ROV surveys (discussed in *Everett & Park, 2018*). The lack of easily identifiable diagnostic features from video, rugose terrain (e.g., overhangs and ledges), and rarity make most smaller species difficult to identify from ROV surveys without voucher specimens. A more thorough analysis incorporating modern (e.g., higher-resolution ROV video surveys with in situ collections) and historical datasets (e.g., museum-archived specimens) would be helpful in elucidating biogeographic patterns more broadly across the basin.

Several species in the genus *Crypthelia* occur in the eastern Caribbean at bathyal depths (*Cairns, 1986*). The difficulty of identifying stylasterids from ROV video remains a challenge in establishing species occurrences, necessitating a voucher collection to confirm morphospecies identity. Nevertheless, the depth distribution, colony morphology, and coloration of *Stylaster* sp. 1, is consistent with *Stylaster roseus* (Pallas, 1766), a tropically-distributed western Atlantic stylasterid from depths typically less than 500 m (*Ocean Biogeographic Information System (OBIS), 2019*). *Crypthelia* sp. and other stylasterid species are more likely to be underreported due to their small size (2–5 cm), on the threshold of being able to be accurately recorded from ROV video. These stylasterids were also observed under overhangs which made obtaining an accurate account of their abundance difficult. In this case, the reported occurrences of these corals represent a minimum value.

The ecological contribution of stylasterid corals is often understated, despite the ability of some species to produce significant three-dimensional structures that can act as habitat for other organisms. Larger structure-forming stylasterids are functionally analogous to some colonial scleractinian corals in that they can provide habitat for larger organisms such as deep-water fishes and invertebrates (*Love, Lenarz & Snook, 2010*; *Braga-Henriques et al., 2011*). *Crypthelia* spp., while relatively abundant in places and occurring over a wide range, were never observed above 12 cm in overall height and were observed to be extremely brittle when attempting collection (Fig. 9A). However, two species in the genus *Stylaster*, *S.* cf. *duchassaingi* (Fig. 9B) and *Stylaster* sp. 1 (Fig. 9C), can be classified as potentially important three-dimensional structure-forming species, primarily occurring between 150 and 400 m depth.

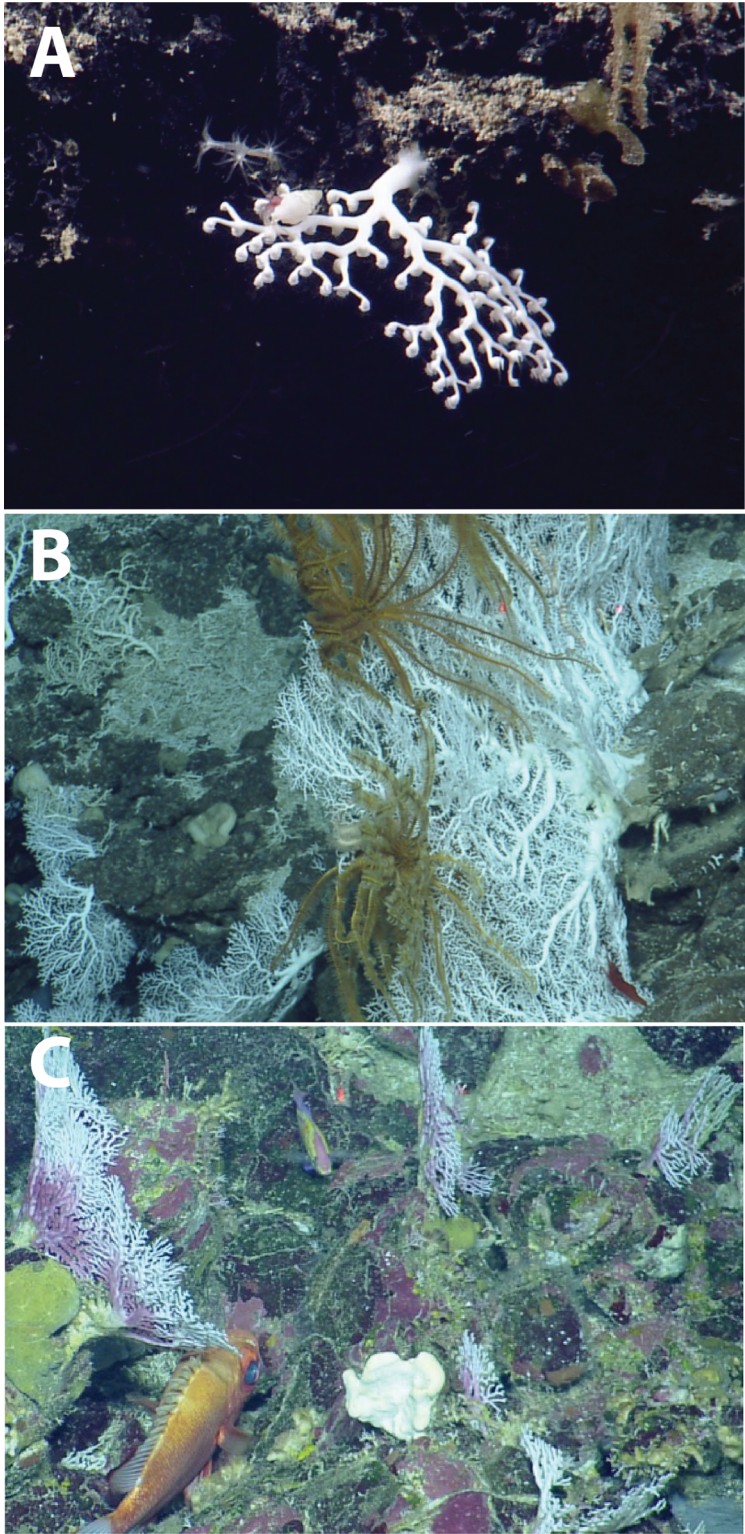

**Figure 9 Deep-water stylasterids from the Anegada Passage seamounts.** (A) *Crypthelia* sp. 1.
(B) *Stylaster* cf. *duchassaingi*. (C) *Stylaster* sp. 1.

The carbonate mineralogy of most stylasterids is similar to scleractinians in that a majority of known species, including those observed in this study, produce a skeleton composed primarily of the mineral aragonite (*Cairns, 2011*), although some are calcitic in composition (*Cairns & Macintyre, 1992*). Stylasterids do share distribution characteristics and overlapping depth ranges with many upper bathyal solitary and colonial scleractinian corals (*Cairns, 1986*, *2011*). Like scleractinians, stylasterid hydrocorals also exhibit a vulnerability to changing aragonite saturation conditions over their depth distribution (*Guinotte et al., 2006*).

Knowledge of the chemical environment of deep-sea scleractinians has grown significantly in recent years as empirical measurements of the carbonate system at deep-sea coral habitats have been reported from different regions across the global ocean. On seamounts in the Indian Ocean off the coast of SW Australia, the majority of framework-forming scleractinians—including *E. rostrata* and *S. variabilis*—were found at $\Omega_{arag}$ values at or just below saturation, suggesting a control on the lower limit of these species' distributions (*Thresher et al., 2011*). However, live scleractinian reefs were recently discovered on seamounts in the North Pacific at $\Omega_{arag}$ values as low as 0.71 (*Baco et al., 2017*) and as low as 0.81 off Southern California (*Gómez et al., 2018*). Additionally, scleractinians have been observed below the ASH in the Indian Ocean (*Trotter et al., 2019*) and South Atlantic Ocean (*Barbosa, Davies & Sumida, 2020*). These revelations indicate that, under the right conditions, scleractinian corals can persist in undersaturated waters (*Baco et al., 2017*). Additionally, the presence of live tissue may buffer against the effects of low pH (*Venn et al., 2011*), but the underlying dead coral framework may be less resilient to undersaturated waters. Expanded surveys below 2,000 m in northern Caribbean, in tandem with additional effort in sampling of coral-adjacent deep-waters for carbonate chemistry analysis will aid in better determining the controls on aragonitic coral species distribution at lower bathyal depths in the tropical western Atlantic.

Productivity of the surface waters and export to the deep-sea benthos may contribute to differences in species distribution and abundance of deep-sea corals. In a 2016 laboratory experiment and field study comparing two spatially distinct and genetically isolated populations of the framework-forming scleractinian *Lophelia pertusa*, colonies from Norway exhibited enhanced respiration and prey capture rates under acidified conditions compared to individuals from the Gulf of Mexico (*Georgian et al., 2016b*). This study lends support to the hypothesis that species are locally adapted to environmental conditions, including food supply, which may allow individuals to better tolerate reduced carbonate saturation states.

## CONCLUSIONS

Our results offer some insights to the distribution, diversity, and drivers of community assembly of scleractinian and stylasterid deep-water corals in a data-deficient region of the tropical western Atlantic Ocean. These findings lend support for the efficacy of targeted exploration surveys to understand coral distribution with respect to environmental

gradients in the deep-sea environment. The presence of aragonitic corals, largely occurring above the aragonite saturation horizon, was not unexpected, but more surprising was the presence of known framework-forming scleractinians, like *Madrepora oculata*, living at or just above $\Omega_{\text{arag}} = 1$ in this locality. Future work should seek to expand upon coral community inventories for this area to include members of the Octocorallia and Antipatharia, particularly to refine the relationship between community assembly and water mass structure. Increased taxonomic effort is needed to better identify cryptic coral morphospecies from the deep-sea benthos, and particularly for those size classes below the identification threshold for ROV video surveys. In the Atlantic Ocean, deep-water coral ecosystem health is likely to be negatively impacted by environmental change by the end of the current century. Warming deep-waters (*Gleckler et al., 2016*), thermohaline driven deep-water acidification (*Pérez et al., 2018*), and low latitude deoxygenation in the Atlantic (*Montes et al., 2016*) are among the greatest threats facing deep-water framework-forming corals. The relationship between coral occurrences and environmental variables reported here establish a critical baseline for the detection of the effects of deep ocean change to seafloor communities, which is crucial to effective conservation.

## ACKNOWLEDGEMENTS

We would like to thank the efforts of the science party, master, and crew of the E/V *Nautilus* during the *Exploration of the Anegada Passage* (NA052) cruise. In addition, we would like to extend a special thanks to the Ocean Exploration Trust for their support. Taxonomic assistance in identifying voucher material was provided by Stephen Cairns (National Museum of Natural History, Smithsonian Institution) for scleractinian and stylasterid corals. We would also like to thank Jason Chaytor, Alex Rogers, Iván Hernández-Ávila, and Nadia Santodomingo for their feedback in the revision of this manuscript. Any use of trade, firm, or product names is for descriptive purposes only and does not imply endorsement by the U.S. Government.

### Funding

Funding for this work was provided to Amanda Demopoulos through an Interagency Agreement (IAA) (OER14-001) with NOAA Office of Ocean Exploration and Research from this IAA. Funding for Erik Cordes was provided through a cooperative agreement between USGS and Temple University. The funders had no role in study design, data collection and analysis, decision to publish, or preparation of the manuscript.

### Grant Disclosures

The following grant information was disclosed by the authors:
Interagency Agreement (IAA): OER14-001.
USGS and Temple University.

## Competing Interests

Erik E. Cordes is an Academic Editor for PeerJ.

## Author Contributions

- Steven R. Auscavitch conceived and designed the experiments, performed the experiments, analyzed the data, prepared figures and/or tables, authored or reviewed drafts of the paper, and approved the final draft.
- Jay J. Lunden conceived and designed the experiments, performed the experiments, analyzed the data, prepared figures and/or tables, authored or reviewed drafts of the paper, and approved the final draft.
- Alexandria Barkman analyzed the data, authored or reviewed drafts of the paper, and approved the final draft.
- Andrea M. Quattrini analyzed the data, authored or reviewed drafts of the paper, and approved the final draft.
- Amanda W.J. Demopoulos conceived and designed the experiments, analyzed the data, authored or reviewed drafts of the paper, and approved the final draft.
- Erik E. Cordes conceived and designed the experiments, analyzed the data, authored or reviewed drafts of the paper, and approved the final draft.

## Data Availability

Raw videos are available from the Ocean Exploration Trust at http://www.oceanexplorationtrust.org/data-request.

The species occurrence data is available as a Supplemental File and is archived for public distribution through the NOAA Deep-sea Coral Database.

https://www.ncei.noaa.gov/maps/deep-sea-corals/mapSites.htm.

## Supplemental Information

Supplemental information for this article can be found online at http://dx.doi.org/10.7717/peerj.9523#supplemental-information.

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
