# Peer review of "Distribution of deep-water scleractinian and stylasterid corals across abiotic environmental gradients on three seamounts in the Anegada Passage"

_PeerJ, doi:10.7717/peerj.9523_

## Round 0.1 · original submission · Minor Revisions

Dear Steven and co-authors,

I have now received two very positive reviews (well done), with a third reviewer due 10 days ago and not responding anymore. Therefore, I will base my decision on the two available reviews. Both reviewers recognised the value, quality and novelty of your work, but have also indicated a number of minor issues that you need to address. For example, Reviewer #1 questioned the validity of your experimental design in measuring the influence of aragonite on corals. The reviewer suggests that you reframe your introduction and discussion accordingly. Reviewer #2 noted some statistical inconsistencies that need fixing (see the provided annotated pdf with comments).

I will be looking forward receiving your revised manuscript along with point-by-point responses to their comments.

With warm regards,
Xavier

·

Basic reporting

In general the paper is well written with clear English. Overall the references are fine. Lines 62-65 could have done with a few more references. Sufficient background and context are also given in the paper. The structure of the paper is also fine and figures / tables etc. seem appropriate.

In the introduction two main reasons are given for the study, the first is that the region studied the Anegada Passage, an important link in deep waters between the Atlantic and Caribbean, is poorly understood in terms of coral distribution. Producing better maps of these habitats as well as ground-truthing of habitat suitability modelling of coral distribution requires more local data from areas like this. This is a straightforward and worthy goal of the paper. The second goal seems to be to see if the corals are influenced by aragonite saturation levels in the surrounding waters. However, only one ROV transect appears to sample water undersaturated with aragonite so I am not convinced the sampling design is up to testing such as question. However, the aragonite saturation measurements are relevant in a wider context and should be retained in the paper. This will require a litte reframing in the introduction and discussion sections of the paper.

Experimental design

The paper fits within the scope of PeerJ. As stated above, however, whilst one of the research questions is well defined the other does not really stand up to the sampling that was achieved during the research expedition. Aragonite data are interesting with respect to coral data regardless, but they cannot be used to infer a limitation of cold-water corals below the aragonite saturation horizon in this case, not enough samples / transects at these depths. The investigation in general terms uses appropriate methods and these are well described and the study could be replicated.

Validity of the findings

The results presented here are novel in terms of providing coral occurrence data from seamounts in the Anageda Passage in the Caribbean. At a regional and global scale the data are incremental but as the study states such data are nonetheless important in filling in the picture of cold-water coral occurrence in the deep sea. The underlying data are provided. There are a few issues with the analyses.

L284-286 - There seems to be something wrong with the p-values here, may be switched around?
L 286-L288 The ANOSIM carried out here between water masses revealed significant differences in assemblages. However, surely these results are confounded by the fact that water mass varies by depth and depth is a well-known driver of coral distribution, and indeed the distribution of rariphotic and bathyal taxa in general. Is it possible to block for depth as an influencing factor in this analysis and undertake an ANOVA or ANCOVA? Or analyse the influence of depth and block for water mass.

L340 As I stated above sampling was so limited in waters unsaturated for aragonite (omega below 1) that it is difficult to draw conclusions about coral distribution with respect to the ASH depth in this study.

L384-386 If species are rarely observed then sampling may not pick them up from some seamounts.

The results and discussion should be modified to account for these comments.

Additional comments

Specific comments:

What are high profile seamounts, are they those with a high elevation or particularly steep seamounts?

L114 E/V Nautilus should be in italics.
L131 ROV Hercules should be in italics.

L157-158 I would be hesitant about saying that stylasterids are functionally similar to framework building corals in their ability to provide habitat for other species. May be so in terms of epizoic and endozoic species but not in terms of the wide variety of other habitats framework building coral provide such as rubble and sediment habitats. They are more like gorgonians which provide habitat in the deep sea in coral garden habitats.

L209 Only 22 observations of solitary corals but of course they are hard to see in an ROV video.

L372 Madracis myriaster should be in italics.

L446 Madrepora oculata is not a primary framework building coral but is a secondary framework forming coral.

·

Basic reporting

The manuscript from Auscavith et al. is a well written solid paper. It bring very good information about the structure of coral assemblages at the Anegada Passage. In particular the approach of consider the Omega Arg estimation associated with assemblage composition was very useful for future approaches on deep-sea distribution of coral assemblages. The literature references are well managed, with few recommendation to add. In general the structure of the article is good, as well as the figures, tables and raw data.
The data obtained and the statistical analyses are fine and include a nice discussion concerning their data output and the need of future research.

Experimental design

The scientific question of the study is well defined and the collected samples and data treatment are allows to address the question. Also the topic is very much relevant and contribute to information about Omega Arg in the Caribbean and their relation with coral composition. Technical and ethical standards seem fulfilled.
On the results section, small inconsistences were detected between a couple of p-values and the occurrence of significant differences (according the text). However, after revision of supplementary information the mistake was evident, and is noted on the pdf file.

Validity of the findings

Data and statistical analyses looks robust and ready for plublication after few amendments of some p-values. Conclusion are well stated, pertinent and supported by the findings.
There is some comments and suggestion on the discussion section in order to complement some comments. Other than that, look fine and interesting.

Additional comments

Congratulation of the paper, it is a very good jod. I look foward your publication. Please check some coments and corrections included in the manucrit file.

Reviewer 3 ·

Basic reporting

See annotated manuscript

Experimental design

See annotated manuscript

Validity of the findings

See annotated manuscript

Additional comments

See annotated manuscript

Annotated reviews are not available for download in order to protect the identity of reviewers who chose to remain anonymous.

---

## Round 0.2 · Minor Revisions

Dear Steven and colleagues,

I have received two final reviews of your revised manuscript, both suggesting some very minor changes. Please not that reviewer 1 has provided you with comments in an annotated copy of the ms. I have also reviewed the ms and have highlighted some minor typos for you to correct (see editor_annotated pdf) - please pay particular attention to the Reference list, which needs improvements.

I will be looking forward to receiving your final manuscript.

With warm regards,
Xavier

·

Basic reporting

The paper has been improved with revision including in terms of clarity, use of references and tidying up some minor errors. I note that the authors have missed the reference Stefanoudis et al (2019) with reference to the depth distribution of Madracis myriaster. I've added a note on the attached ms. I also highlight a sentence in the ms which I do not think is a reasonable one to write on the basis of the results.

Experimental design

This is fine. The authors have defended their analysis adequately in the rebuttal.

Validity of the findings

The adjustment of the language has clarified the findings with respect to the aragonite saturation horizon. Note the highlighted sentence referred to above.

Additional comments

The paper requires some further minor edits but I do not need to see it again. Many thanks for responding to my comments.

Reviewer 3 ·

Basic reporting

Dear Editor and authors,
Thanks for your invitation to review the updated version of the manuscript by Auscavitch and collaborators. This study presents an excellent contribution to the knowledge of deep-water corals (scleractinians and stylasterids) in the western Atlantic. The study provides a valuable baseline to understand environmental factors (omega-aragonite saturation values related to depth and temperature) that control the distribution of deep-sea corals in this biogeographic region. This information is critical to understand the future of deep-water fauna under different climate change scenarios.

My only additional comment is that the authors keep the use of Lophelia pertusa instead of the valid name Desmophyllum pertusum (Worms: http://www.marinespecies.org/aphia.php?p=taxdetails&id=1245747). I understand that Lophelia has been extensively used in the literature and the name is attached to the memory of a wide scientific community. However, I encourage the authors to follow the taxonomic nomenclature rules of ICZN as they have properly used in the identification of the material in their manuscript.

I strongly support the publication of this manuscript in this journal.

Experimental design

The authors have incorporated the changes requested by the reviewers. They also answered and justified their choice for the statistical tests used.
I acknowledge that the authors provided a list of the voucher specimens they used in their analyses.

Validity of the findings

The manuscript has a high standard in the amount of information provided, methods, quality of figures, and analyses that support their conclusions.

Additional comments

Congratulations to the authors. This study enhances our knowledge in the ecology of deep-sea corals in the region.

I look forward to your publication.

---

## Round 0.3 · accepted · Accept

I apologize that Xavier Pochon is not available right now, so I have stepped in to complete the editorial process for your manuscript. I have now read all the past reviews, your revisions, and the final version of the currently submitted manuscript.

I pondered your hesitation to change the taxonomic name of Lophelia to Desmophyllum as requested by reviewers. While it is journal policy to use the currently accepted taxonomic names, I also note that the WoRMS entry includes dissention by Cairns on this species being renamed based on available evidence. Given that you are referring to the published name from previous work rather than reporting on this species directly in your manuscript, I am content to accept your decision to follow Cairns in waiting to see what happens with the taxonomy of this species.

This is the only discrepancy between the revised manuscript and the recommendations of the referees, so I see no reason to delay the process further. I am satisfied that your manuscript is acceptable for publication and am happy to move it forward into production.